# Antimicrobial Peptides SET-M33L and SET-M33L-PEG Are Promising Agents Against Strong Biofilm-Forming *P. aeruginosa*, Including Multidrug-Resistant Isolates

**DOI:** 10.3390/antibiotics14070699

**Published:** 2025-07-11

**Authors:** Alessio Fontanot, Peter D. Croughs, Clelia Cortese, Adrianus C. J. M. de Bruijn, Chiara Falciani, Alessandro Pini, Isabella Ellinger, Wendy W. J. Unger, John P. Hays

**Affiliations:** 1Laboratory of Pediatrics, Department of Pediatrics, Sophia Children’s Hospital, Erasmus University Medical Center (Erasmus MC), 3015 GD Rotterdam, The Netherlands; a.fontanot@erasmusmc.nl (A.F.); a.c.j.m.debruijn@erasmusmc.nl (A.C.J.M.d.B.); w.unger@erasmusmc.nl (W.W.J.U.); 2Department of Medical Microbiology & Infectious Diseases, Erasmus University Medical Center (Erasmus MC), 3015 GD Rotterdam, The Netherlands; p.croughs@erasmusmc.nl; 3Institute of Pathophysiology and Allergy Research, Center for Pathophysiology, Infectiology and Immunology, Medical University of Vienna, 1090 Vienna, Austria; isabella.ellinger@meduniwien.ac.at; 4Department of Medical Biotechnology, University of Siena, via A. Moro 2, 53100 Siena, Italy; clelia.cortese@student.unisi.it (C.C.); chiara.falciani@unisi.it (C.F.); alessandro.pini@unisi.it (A.P.); 5SetLance srl, via Fiorentina 1, 53100 Siena, Italy; 6Clinical Pathology Unit, Santa Maria alle Scotte Hospital, Viale M. Bracci, 53100 Siena, Italy

**Keywords:** *Pseudomonas aeruginosa*, biofilm, antimicrobial peptides, multidrug resistance, SET-M33L, SET-M33L-PEG

## Abstract

**Background**: The antimicrobial peptides (AMPs) SET-M33L and SET-M33L-PEG were investigated against 10 clinical isolates of *P. aeruginosa*. **Methods**: Their minimum inhibitory concentrations (MICs), minimum bactericidal concentrations (MBCs), and minimum biofilm inhibitory concentrations (MBICs) were evaluated against tobramycin, ceftazidime, and polymyxin B. **Results**: MICs and MBCs were 7- to 100-fold lower than tobramycin, and 10- to 300-fold lower than ceftazidime. Fractional inhibitory concentration (FIC) indices showed an additive effect, while fractional bactericidal concentration (FBC) indices showed synergistic effects (FBC < 0.5) for most isolates. **Conclusion**: SET-M33L and SET-M33L-PEG are promising antimicrobial agents against strong biofilm-forming *P. aeruginosa*, including MDR isolates.

## 1. Introduction

The continuing increase in the prevalence of antimicrobial resistance (AMR) is a significant health threat that seriously impacts on the efficacy of global antimicrobial therapies. For example, in 2019 alone, AMR caused an estimated 4.95 million deaths worldwide [1]. Additionally, research showed that only six pathogens (*Pseudomonas aeruginosa*, *Staphylococcus aureus*, *Klebsiella pneumoniae*, *Streptococcus pneumoniae*, *Acinetobacter baumannii*, and *Escherichia coli*), were responsible for 929,000 out of 1.27 million deaths directly attributed to AMR bacteria, causing significant morbidity, mortality, and financial burdens to healthcare systems globally [2]. AMR is also one of the most frequent factors that hinder the effective treatment of pulmonary infections, especially infections associated with chronic lung diseases such as bronchiectasis and cystic fibrosis (CF) [3,4]. CF patients in particular suffer from frequent and persistent lung infections, most commonly caused by *P. aeruginosa*. The bacterium has the ability to secrete a variety of virulence factors that facilitate tissue invasion and trigger chronic inflammation. *P. aeruginosa* is also able to produce extracellular polymeric substances that help it to form complex and difficult-to-treat biofilms, which along with various intrinsic multi-drug resistance mechanisms, enhance the bacterium’s ability to resist many different types of antibiotics [4,5,6,7]. Biofilms are complex communities of microorganisms that adhere to surfaces and form protective structures via the production of an extracellular matrix (ECM) in which the bacterial cells are embedded. These structures contribute to chronic infections and AMR in clinical settings [8,9]. Distinctive features of biofilms include ECM exopolysaccharide and protein components, which act to hinder the diffusion of antimicrobials into metabolically inactive bacterial “persister” cells, and quorum-sensing mechanisms that modulate gene expression [10]. Biofilm formation is often associated with the use of implants, biomaterials, and medical devices in patients, which create favorable conditions for microbial colonization and difficult-to-treat infections [8].

Antimicrobial peptides (AMPs) are a promising alternative approach to combating multidrug-resistant (MDR) bacteria and biofilm formation [11]. These small molecules have demonstrated wide-ranging antibacterial activity related to their cationic charge, beta-fold formation, amphipathic properties, and hydrophobicity [12]. Several thousand AMPs have been described [13], being able to act on multiple targets present in/on pathogens, with the high number of recent developments in this field indicating the potential of these antimicrobial compounds.

SET-M33L is an L-amino acid synthetic tetra-branched AMP that has previously been shown to exhibit enhanced stability and bioavailability in biological fluids when compared to the linear version of this AMP molecule, as well as only generating mild signs of toxicity (disappearing in a few hours) [14]. SET-M33L also showed antimicrobial activity against a range of MDR Gram-negative bacteria, including *P. aeruginosa*, *E. coli*, *A. baumannii*, and *K. pneumoniae* [15,16]. Its mechanism of action is based on a two-step process: high-affinity binding to lipopolysaccharide, followed by a disruption of the bacterial membrane [16,17]. SET-M33L showed minimum inhibitory concentration (MIC) values between 1.5 μM and 3 μM against standard strains of *P. aeruginosa* [18]. Subsequent PEGylation of the C-terminal threonyl branch to yield SET-M33L-PEG (see Appendix A for structural information) was shown to enhance the AMP’s stability against *P. aeruginosa* elastase, an important virulence factor that could possibly degrade the peptide [19]. Previous research using these SET-M33 AMPs was based on planktonic bacteria, with further studies on the efficacy of SET-M33L and SET-M33L-PEG against MDR and biofilm-forming clinical isolates of *P. aeruginosa* being warranted. Therefore, the current research investigated the in vitro antimicrobial activity of these two AMPs on a range of MDR, ‘strong’ biofilm-forming *P. aeruginosa* isolates from patients presenting with chronic pulmonary disease, including bronchiectasis and CF.

## 2. Results

Initial experiments were performed on a diverse set of 17 biofilm-forming clinical isolates of *P. aeruginosa*, so that a set of strong biofilm-forming isolates (MDR and non-MDR variants) could be selected for further analysis using SET-M33L and SET-M33L-PEG AMPs.

### 2.1. Antibiotic Susceptibility Profiles of Clinical P. aeruginosa Isolates

Antimicrobial susceptibility testing of 17 clinical isolates of *P. aeruginosa* against nine conventional antimicrobials indicated varying levels of resistance/susceptibility (Table 1). A significant majority of the isolates (12/17, 71%) exhibited resistance to at least one antimicrobial, while nearly one-third (6/17, 35%) were classified as resistant to five or more of the antimicrobials tested on that specific isolate and found to be MDR isolates, i.e., resistant to one or more antimicrobials in three different classes [20]. The variable range of the results highlighted the complex panel of resistance seen in chronic *P. aeruginosa*-mediated infections in pulmonary disease patients.

Table 1 shows the antimicrobial susceptibility of 17 clinical *P. aeruginosa* isolates to nine antimicrobials according to clinical breakpoints by last EUCAST guidelines (v15). S = susceptible, standard dosing regimen; I = susceptible, increased exposure; R = resistant; nt = not tested. MIC and antimicrobial sensitivity values were obtained using Vitek2 and disk diffusion testing, except for colistin values, which were obtained using microdilution testing. Antimicrobials tested were aminoglycosides (AMG)—tobramycin (TOB) and gentamicin (GEN); beta-lactams (BL)—ceftazidime (CAZ), piperacillin–tazobactam (PIP-TAZ), and aztreonam (AZT); polymyxins (PMX)—colistin (COL); carbapenems (CPN)—meropenem (MER) and imipenem (IMP); and fluoroquinolones (FQ)—ciprofloxacin (CIP). IE indicates that there is insufficient evidence that the *P. aeruginosa* is a good target for therapy with this antimicrobial, as described in the latest EUCAST guidelines (v15) document. Classifications in brackets are used as described in EUCAST’s guidance document, “When there are no breakpoints” (3 September 2024), according to the following: “comparing the MIC to MIC distributions on http://mic.eucast.org or in the literature: if the MIC was above the ECOFF (and thus higher than MICs of the wild-type distribution), clinical use of the antibiotic has been reported as “resistant”. If MIC was below or equal to the ECOFF, clinical use can be considered, but has not been reported as “susceptible”. Background color indicates clustering of different antibiotic classes. Bold text highlights resistant isolates.

### 2.2. Biofilm-Forming Ability

A total of 10 of the 17 isolates (10/17, 59%) exhibited ‘strong’ biofilm-forming ability compared to the standard strain PAO1, with the average OD550 of eluted crystal violet exceeding an OD550 of 1.35, i.e., >50% of the average of PAO1 (OD550 of 2.7). These results were used alongside the results on antimicrobial sensitivity testing (Section 2.1) to determine which of the 10 strong biofilm-forming isolates were MDR or non-MDR (Figure 1). The final selection included two susceptible strains (27547, 31008); four resistant strains, which were resistant only to one or two antimicrobial classes (2002, 72774, 30375, 30229); and four MDR strains, which were resistant to three or more antimicrobial classes (27369, 30475, 27039, 30429). All biofilm experiments were conducted using the standard strain PAO1 as a control [21].

### 2.3. Minimum Inhibitory Concentration (MIC) of AMPs

MICs and MBCs of SET-M33L and SET-M33L-PEG were determined against the 10 selected clinical isolates of *P. aeruginosa* and comparator strain PAO1. Two conventional antimicrobials were used as controls: tobramycin (an aminoglycoside) and ceftazidime (a third-generation cephalosporin). These antimicrobials were chosen due to their frequent use as first-line antimicrobials for treating pulmonary *P. aeruginosa* infections [21,22].

All the results and considerations about the AMPs tested are stated in µM to standardize comparisons across a wide range of peptides with varied molecular weights, thereby ensuring consistency in the evaluation of antimicrobial activities (Table 2). Results expressed in µg/mL are available in the Appendix A).

The two SET-M33L compounds successfully inhibited the planktonic growth of all 10 clinical isolates, with MICs values ranging between 0.3 μM and 10.9 μM. For the two conventional antimicrobials, MICs ranged between 1.8 μM and 58.5 μM for ceftazidime and between 2.1 μM and 547.6 μM for tobramycin. For the comparator strain PAO1, SET-M33 MIC values were 2.7 μM, but 3.7 μM for ceftazidime and 4.3 μM for tobramycin. Overall, MICs for the AMPs were 6-fold lower than ceftazidime and 20-fold lower than tobramycin.

A similar trend was observed for the bactericidal analysis, with MBC values ranging between 0.7 µM and 43.7 µM for the AMPs, but between 3.7 µM and 234.2 µM for ceftazidime, and between 8.5 µM and 1095.2 µM for tobramycin. In general, the MBC for the AMPs was 7-fold lower than ceftazidime and 17-fold lower than tobramycin. Interestingly, for the comparator strain PAO1, the opposite phenomenon was observed, with MBC values being 3.7 μM for ceftazidime, 8.5 μM for tobramycin, 10.6 μM for SET-M33L-PEG, and 10.9 μM for SET-M33L.

Overall, the SET-M33 AMPs displayed antimicrobial efficacy on all of the MDR strains (27369, 30475, 27039, 30429) at lower concentrations than the two conventional antimicrobials. In tobramycin-resistant strains (27039, 30429), AMPs’ MICs and MBCs were 7- to 100-fold lower than those of tobramycin. In ceftazidime-resistant strains (27369, 30475, 27039), AMPs’ MICs and MBCs were 10- to 300-fold lower compared to ceftazidime.

Finally, we observed that SET-M33L-PEG displayed both lower MICs and MBCs compared to SET-M33L in 8/10 (80%) of the clinical isolates.

### 2.4. Minimum Biofilm Inhibitory Concentration (MBIC)

The MBIC of M33L and SET-M33L-PEG was determined against the 10 selected isolates of *P. aeruginosa* and standard strain PAO1, using the conventional antimicrobial tobramycin and clinically applied AMP polymyxin B as conventional antimicrobials. The results showed that the AMPs inhibited biofilm formation in all selected clinical isolates at concentrations ranging from 0.3 to 21.8 µM (Table 3 (µM), Appendix A (µg/mL) and Appendix A). For the comparator strain PAO1, these values ranged from 2.7 to 5.5 µM, while for the conventional antimicrobials, tobramycin and polymyxin B, values ranged between 2.1 and 273.8 µM and 0.2 and 3.3 µM, respectively. However, one clinical isolate (30375) showed very high resistance to polymyxin B, with an MBIC > 106.3 µM. When focusing on MDR isolates, AMPs’ MBICs ranged from 0.3 to 21.8 µM, while non-MDR isolates ranged from 0.6 to 21.8 µM. Essentially, a lower concentration of one or both AMPs was required to inhibit biofilm formation for both MDR and non-MDR isolates when compared to at least one of the two conventional antimicrobials used, although polymyxin B showed lower MBIC values compared to the SET-M33L AMP. Finally, SET-M33L-PEG displayed lower or similar MBICs compared to SET-M33L in 9/10 (90%) of isolates.

### 2.5. Bacterial Viability in Biofilms After MBIC Testing

The viability of *P. aeruginosa* isolates in biofilms after MBIC testing was assessed by colony counting. A reduction of over 90% in viability compared to the respective untreated controls was observed across all the investigated strains, ranging from 0.5xMBIC to 2xMBIC for all antimicrobials (Table 4 (µM) and Appendix A (µg/mL)). Notably, in this concentration ranges, the two SET-M33 AMPs showed a complete killing (100% viability reduction—no regrowth) in half of the strains (5/10). Specifically, they displayed this result in 2/4 of the MDR isolates, 3/6 of the non-MDR isolates, and in the comparator strain PAO1. No re-growth was observed in over two-thirds (7/10) of the isolates for tobramycin. Finally, although Polymyxin B exhibited the lowest MBICs, complete killing within this concentration range was observed in less than one-third (3/10) of the isolates.

### 2.6. Antimicrobial Synergy—Fractional Inhibitory Concentration (FIC) and Fractional Bactericidal Concentration (FBC)

Potential synergy of the AMPs with ceftazidime and tobramycin on the MIC and MBC of clinical *P. aeruginosa* isolates was assessed by calculating the FIC (Table 5) and FBC (Table 6) indices. For FIC, synergy between SET-M33L or SET-M33L-PEG with ceftazidime was observed in two different isolates (30229 and 27039, respectively) as determined by a FIC ≤ 0.5. Interestingly, isolate 27039 is an MDR strain for both tobramycin and ceftazidime. No synergy was observed between SET-M33 AMPs and tobramycin for the FIC indices. In fact, most FIC interactions were ‘additive’ (22/40, 55%), with a FIC index between 0.5 and 1. The remaining interactions, 16/40 (40%), were ‘indifferent’ (1 < FIC index ≤ 4). Additive effects were primarily observed when higher concentrations of antimicrobials (yet still below their MICs) were combined with varying AMP concentrations. Additionally, SET-M33L displayed additive effects against more isolates than SET-M33L-PEG (8/10, 80% versus 6/10, 60%). No combinations of AMPs and conventional antibiotics generated antagonism (FIC > 4).

For FBC, synergy (FBC ≤ 0.5) between SET-M33L and tobramycin (isolates 27547, 30375 and 30229), and between SET-M33L and ceftazidime (isolates 31008, 30375, 30429 and 30229) were observed. Synergy between SET-M33L-PEG with tobramycin (isolates 27369, 31008 and 30375) and with ceftazidime (isolates 27547, 27039, 31008, 30375, 30429 and 30229) was observed. However, the majority of FBC interactions (21/40, 53%) between the AMPs and tobramycin or ceftazidime were additive (0.5 < FBC index ≤ 1), with the remaining interactions (3/40, 8%) being indifferent (1 < FBC index ≤ 4), isolates). No combinations showed antagonism (FBC > 4). When using comparator strain PAO1, combinations of AMPs and conventional antimicrobials generated additive effects for both FIC and FBC indices.

## 3. Discussion

A selection of 10 *P. aeruginosa* clinical isolates with strong biofilm-forming ability and a diverse pattern of resistance/susceptibility to conventional antimicrobials was established [6] and tested against SET-M33L and SET-M33L-PEG. The results demonstrated lower MICs, MBCs, and MBICs for SET-M33L and SET-M33L-PEG (as measured in µM concentrations) when compared to tobramycin and ceftazidime. In general, the PEGylated version of the AMP was effective at lower concentrations than the non-PEGylated version. Additionally, both AMPs tended to show additive bacteriostatic effects and synergistic bactericidal effects when combined with the conventional antibiotics, tobramycin and ceftazidime. Our data also revealed that both AMPs possessed promising antimicrobial activity against clinical *P. aeruginosa* isolates obtained from difficult-to-treat infections, including all of the MDR isolates tested. Notably, in tobramycin-resistant isolates, AMPs’ MICs and MBCs values were 7- to 100-fold lower compared to tobramycin, while in ceftazidime-resistant strains, these values were 10- to 300-fold lower compared to ceftazidime. The higher sensitivity of the tobramycin- and ceftazidime-resistant isolates for the SET-M33 AMPs may provide alternative treatment options, which may result in faster clearance of the resistant isolates than when using the conventional antimicrobials. This could impact on disease duration lowering, the chance of treatment failure, and prolonged illness [23]. The superior antimicrobial performance of SET-M33 AMPs over conventional antimicrobials could be attributed to their tetrameric conformation, which offers enhanced protection against protease degradation.

With respect to biofilm inhibition, SET-M33 AMPs outperformed the conventional antimicrobial tobramycin in 9/10 of the isolates, including one of the two tobramycin-resistant strains and two MDR isolates. Yet, in turn, the SET-M33 AMPs were outperformed by the conventional antimicrobial peptide polymyxin B in 9/10 isolates. However, while SET-M33 AMPs achieved a complete killing (i.e., a 100% reduction in viability) in half (5/10) of the isolates at the concentration range from 1xMBIC to 2xMBIC, polymyxin B only demonstrated 100% killing in less than one third (3/10) of isolates. However, the reduced number of isolates 100% killed by polymyxin B at its 1xMBIC or 2xMBIC indicates a risk of outgrowth of residual bacteria following polymyxin B treatment. This suggests that treatment using SET-M33 AMPs at its MBIC or 2xMBIC could potentially decrease the risk of re-growth of treated clinical isolates of *P. aeruginosa* during infection. This finding indicates that further research into SET-M33 AMPs pharmacokinetic–pharmacodynamic properties is warranted.

In contrast to SET-M33 AMPs, tobramycin displayed a complete MBIC killing in more than two-thirds of isolates (7/10), but higher concentrations of this antimicrobial were required than of the other compounds. The need for higher concentrations of tobramycin to achieve biofilm inhibition and complete killing could culminate in the induction of resistance in treated strains and in higher toxicity for patients.

Interestingly, when considering biofilm formation and MIC values, SET-M33L-PEG outperformed SET-M33L in 90% of the isolates tested. This finding is encouraging as C-terminal PEGylation potentially improves the pharmacokinetic properties of AMPs by further increasing AMP stability against bacterial proteases, improving solubility and plasma clearance [16,24]. However, there are several disadvantages associated with utilizing AMPs as antimicrobial therapies, including toxicity, stability, limited bioavailability, and production costs [25], although a tetrameric form and PEGylation may be a solution to some of these issues.

The combined use of AMPs, together with conventional antimicrobials, could be a potential method to increase the efficacy of antimicrobial therapy, while reducing the chance of antimicrobial resistance evolving [26]. Although our results showed no synergistic bacteriostatic effects, additive bacteriostatic effects were observed in more than two thirds of the isolates when using SET-M33L, or SET-M33L-PEG, in combination with the antimicrobials tobramycin or ceftazidime. With respect to bactericidal action, both additive and synergistic bactericidal effects were observed (100% of isolates and more than half of the isolates tested, respectively). Interestingly, some combinations of AMPs and conventional antimicrobials generated additive FIC values, but synergistic FBC values. Biologically, this effect may be a consequence of an incomplete understanding of AMP mechanisms of action that possibly involves an interaction with the external membrane, causing increased cell permeabilization [16,17,27]. Such membrane disruption may facilitate the greater penetration of antibiotics into the bacterial cell, thereby facilitating increased antibiotic activity on, for example, antibiotic binding to 30S ribosomal subunits or penicillin-binding proteins [28,29]. These results suggest that AMPs may enhance conventional antimicrobial efficacy at bactericidal concentrations and could potentially increase the efficacy of treating *P. aeruginosa* infections.

## 4. Materials and Methods

### 4.1. M33 Peptides Synthesis

The AMPs tested included SET-M33L and its PEGylated version SET-M33L-PEG. M33 was initially obtained as a trifluoroacetate salt during synthesis and then converted to an acetate form post-synthesis. The peptides were obtained via solid-phase peptide synthesis using 9-fluorenylmethyloxycarbonyl (Fmoc) chemistry. Essentially, Fmoc4-Lys2-Lys-b-Ala Wang resin (Rapp Polymer GmbH, Tuebingen, Germany) was used to construct the L-amino acid version (SET-M33L), while TentaGel^®^ S RAM resin (Rapp Polymer GmbH, Tuebingen, Germany) was used to create the PEGylated form (SET-M33L-PEG), which included Fmoc-NH-Peg4-COOH for PEGylation. Standard groups for serine, arginine, and lysine were used to establish side-chain protection. Following synthesis, both peptides were separated from the resin by cleavage and precipitated utilizing ether, followed by a purification using reversed-phase chromatography, which was verified by mass spectrometry. Peptides were dissolved in MilliQ water and stored at −20 °C [16,19].

### 4.2. Isolation and Characterization of Clinical Isolates of P. aeruginosa

A total of 17 *P. aeruginosa* isolates were used. Isolates were cultured from the sputa of patients presenting with chronic pulmonary diseases (e.g., CF and bronchiectasis), over a period ranging from May 2023 to July 2024, at the Department of Medical Microbiology and Infectious Disease of the Erasmus University Medical Center (Erasmus MC), Rotterdam, The Netherlands (Appendix A). A single isolate from the *P. aeruginosa* ‘Surveillance Collection’ of EMC isolated from a pulmonology patient in 2013 was also included as an antimicrobial susceptible control isolate (A13L9). *P. aeruginosa* standard strain PAO1 was employed as a comparator strain [30]. For isolation, sputum samples were directly plated onto blood chocolate agar plates, and colonies were isolated based on their morphological characteristics. The identification of the isolates was conducted using matrix-assisted laser desorption ionization—time of flight (MALDI-TOF) mass spectrometry (Bruker Daltonics, Billerica, MA, USA). Subsequently, susceptibility testing on a set of 9 conventional antimicrobials (tobramycin, ceftazidime, colistin, meropenem, imipenem, gentamicin, piperacillin–tazobactam, ciprofloxacin, and aztreonam) was performed using the VITEK 2 system (bioMérieux, Marcy-l’Étoile, France) [31]. Disk diffusion testing was performed for difficult-to-grow isolates [32], while for colistin susceptibility, a microdilution method was used [33].

### 4.3. Biofilm-Forming Ability

The biofilm-forming abilities of all 17 *P. aeruginosa* isolates were evaluated under standardized conditions by quantifying biomass using the crystal violet method [34,35] optimized for ibidi 18-well in vitro devices (ibidi GmbH, Grafelfing, Germany). Each experiment was performed in triplicate and with two independent biological replications. The isolates were first grown overnight in 15 mL of fresh Luria–Bertani (LB) medium (Carl Roth GmbH, Karlsruhe, Germany). The next day, bacterial suspensions were homogenized using 0.5 × 16 mm microneedles at 1.5 × 10^8^ CFU/mL. Biofilm cultures were prepared by diluting the suspension 20-fold in a fresh LB medium to obtain a final volume of 100 μL per well in 18-well culture plates, while a negative control consisted of 100 μL of fresh LB medium. Biofilm cultures were then incubated for 24 h at 37 °C in 5% CO_2_. Next, the LB medium was removed, and the wells were washed once with sterile Dulbecco’s phosphate-buffer saline (DPBS: Gibco, Paisley, UK) to remove all non-adherent cells. Biofilms were then fixed with 80 μL of 4% paraformaldehyde (PFA) (Merck KGaA, Darmstadt, Germany) for 30 min at room temperature (RT). PFA was removed and the wells were washed with sterile DPBS. For biomass quantification, biofilms were stained with 80 μL of a solution containing 0.1% crystal violet (Sigma-Aldrich Chemie, Steinheim, Germany) in distilled water, previously filtered through 0.2 μm filters, and incubated at RT for 5 min. The wells were then washed three times with DPBS to remove excess crystal violet and examined using inverted microscopy (DMIL FLUO, Leica Microsystems, Wetzlar, Germany) to assess any distinct morphological features. Finally, 120 μL of 96% ethanol was added to each well for 30 min in order to elute crystal violet from the stained biofilms. The biofilms’ biomass was then quantified by measuring the absorbance of the crystal violet elution at OD550. (SpectraMax iD3). Based on the results obtained, 10 ‘strong’ biofilm-forming isolates were chosen for further AMP testing. Strong biofilm-forming isolates produced an OD550 of at least 50% of the OD550 value obtained for the fully grown biofilm biomass of the standard biofilm-forming *P. aeruginosa* strain PAO1, when quantified under the same conditions (OD550 = 2.7) [36].

### 4.4. Minimum Inhibitory Concentration (MIC) and Minimum Bactericidal Concentration (MBC) of AMPs

The MIC of the two AMPs (SET-M33L and SET-M33L-PEG) and conventional antimicrobials (tobramycin and ceftazidime) against clinical *P. aeruginosa* isolates was assessed according to EUCAST with slight modifications [37,38,39]. MIC values were evaluated on planktonic growth of 10 strong biofilm-forming *P. aeruginosa* clinical isolates, based on previous biomass quantification under standardized conditions (Section 2.3). AMPs and conventional antimicrobials were diluted in a fresh LB medium to a range of concentrations from 0.5 μg/mL to 512 μg/mL [38]. Next, 160 µL was added to each well of a flat-bottom 96-well plate (Greiner Bio-one, Frickenhausen, Germany) containing 40 µL of bacterial inoculum at a concentration of 5 × 10^4^ CFU/mL. The concentrations used were SET-M33L: 0.08–87.4 µM, SET-M33L-PEG: 0.08–84.8 µM, tobramycin: 1.1–1095.2 µM, and ceftazidime: 0.9–936.7 µM. For each isolate, an untreated positive control consisted of 40 μL of bacterial inoculum in 160 μL of fresh LB medium, while the negative control consisted of 200 μL of fresh LB medium. Finally, the plate was incubated at 37 °C in 5% CO_2_, and after 24 h, bacterial growth was assessed by measuring the OD600. Each experiment was performed in triplicate and in two independent biological replications.

For the evaluation of the MBC, the AMPs and the conventional antimicrobials were diluted in a fresh LB medium to a range of concentrations obtained from previous MIC experiments up to 512 μg/mL (SET-M33L: 87.4 µM, SET-M33L-PEG: 84.8 µM, tobramycin: 1095.2 µM, and ceftazidime:936.7 µM). These were then incubated with the planktonic growth of 10 strong biofilm-forming *P. aeruginosa* clinical isolates [37]. After overnight incubation at 37 °C in 5% CO_2_, bacterial suspensions were serially diluted 10-fold, plated out on fresh LB agar, and again incubated for 24–48 h at 37 °C in 5% CO_2_. The MBC was then determined as the point at which no colony growth was observed. Each experiment was performed in triplicate and in two independent biological replications.

### 4.5. Minimum Biofilm Inhibitory Concentration (MBIC)

The inhibitory effect of SET-M33L and SET-M33L-PEG on biofilm formation was assessed using the crystal violet method in 18-well plates, as described in Section 2.3 [34,35,40,41]. AMPs and two conventional antimicrobials (tobramycin [42] and polymyxin B [43]—an AMP already in clinical use) were serially diluted in fresh LB medium at a range of concentrations between 0.2 μg/mL and 128 μg/mL. The concentrations used were SET-M33L: 0.04–21.8 µM, SET-M33L-PEG: 0.04–21.2 µM, tobramycin: 0.5–273.8 µM, and polymyxin B: 0.2–106.4 µM. Then, 95 μL of each dilution was added to 5 μL of *P. aeruginosa* at 1.5 × 10^8^ CFU/mL in each test well. Positive biofilm controls comprised 5 μL of each *P. aeruginosa* isolate in 95 μL of LB medium with a negative control comprising 100 μL of LB medium. After 24 h of incubation at 37 °C in 5% CO_2_, the percentage of biofilm inhibition (measured as change in crystal violet OD550) was determined for each concentration of AMP and conventional antimicrobial and compared to the untreated positive-control biofilm for each isolate (set as 100% biofilm formation for each isolate). MBIC was defined as the concentration that generated at least a 90% reduction in biofilm compared to the untreated control [44]. Each experiment was performed in triplicate and in two independent biological replications.

### 4.6. Bacterial Viability in Biofilms after MBIC Testing

The viability of *P. aeruginosa* isolates was assessed after treatment of biofilms with antimicrobials, as described in Section 2.6, but without adding crystal violet. Instead, after antimicrobial treatment and biofilm formation, the growth medium was removed, and wells were washed with sterile DPBS to remove non-adherent cells. The adherent biofilm was removed by gently scraping the surface of the wells and subsequently resuspended in 200 μL of sterile DPBS. The cell suspension was serially 10-fold diluted in sterile DPBS and 5 μL drops of each dilution were plated in a row on fresh LB agar plates, followed by an incubation for 24 h at 37 °C in 5% CO_2_. Next, the highest dilution that generated visible growth was compared to the highest dilution that generated visible growth in the untreated biofilm. Finally, the CFU/mL was calculated using the highest dilution that generated visible growth, presuming that growth in the highest dilution was equivalent to 1 CFU at the time of culture. Each experiment was performed in triplicate and in two independent biological replications.

### 4.7. Antimicrobial Synergy—Fractional Inhibitory Concentration (FIC) and Fractional Bactericidal Concentration (FBC)

To evaluate the potential inhibitory effect of the two AMPs when used in combination with conventional antimicrobials, the method described in Section 2.4 was used with slight modifications. The 10 clinical *P. aeruginosa* isolates were first cultured overnight in LB broth and then 40 μL from a 5 × 10^4^ CFU/mL solution were pipetted into 96-well flat-bottom plates. Each AMP was then diluted in two-fold concentrations from 0.5 μg/mL to 512 μg/mL and added to corresponding wells. Finally, serial dilutions of tobramycin or ceftazidime were prepared and added in concentrations ranging from MIC to MIC/4, bringing the final volume of each well to 200 μL. After 24 h of incubation at 37 °C in 5% CO_2_, bacterial growth was determined as described in Section 2.4. Potential synergistic interactions between AMPs and the conventional antimicrobials tobramycin or ceftazidime were assessed using the following FIC index calculation [45]:FIC index = FIC (A) + FIC (B),
where

FIC (A) = MIC of drug A in combination/MIC of drug A alone

FIC (B) = MIC of drug B in combination/MIC of drug B alone

The interpretation of the FIC index was performed as follows:(1)FIC index ≤ 0.5: synergy (the drugs work better in combination than alone)(2)0.5 < FIC index ≤ 1: additive (the drugs work equally well in combination and alone)(3)1 < FIC index ≤ 4: indifference (the drugs do not significantly affect each other’s performance)(4)FIC index > 4: antagonism (the drugs work worse in combination than alone)

The FBC index was calculated using an adapted version of the FIC calculation method to evaluate the potential synergistic bactericidal action of the two AMPs when combined with conventional antimicrobials [45]. The bactericidal action was evaluated plating out the treated bacterial suspensions—after 24 h of incubation at 37 °C in 5% CO_2_—on fresh LB agar plates, following the methodology described in Section 2.5. FBC indices were finally calculated as follows:FBC index = FBC (A) + FBC (B),
where

FBC (A) = MBC of drug A in combination/MBC of drug A alone

FBC (B) = MIC of drug B in combination/MIC of drug B alone

The interpretation of the FBC index was performed as follows:(1)FBC index ≤ 0.5: synergy (the drugs work better in combination than alone)(2)0.5 < FBC index ≤ 1: additive (the drugs work equally well in combination as alone)(3)1 < FBC index ≤ 4: indifference (the drugs do not significantly affect each other’s performance)(4)FBC index > 4: antagonism (the drugs work worse in combination than alone)

All FIC and FBC experiments were performed in triplicate and in two independent biological replications.

## 5. Conclusions

This study demonstrated that SET-M33L and SET-M33L-PEG successfully inhibit biofilm formation in clinical isolates of *P. aeruginosa* from patients presenting with pulmonary disease. In MDR isolates, the AMPs demonstrated superior antimicrobial efficacy compared to the conventional ‘first-line’ antimicrobials tobramycin and ceftazidime, which are used in treating CF patients. When tested in combination with the conventional antimicrobials tobramycin and ceftazidime, both SET-M33 peptides displayed additive bacteriostatic and synergistic bactericidal effects against the majority of *P. aeruginosa* isolates, including MDR isolates. Finally, the AMPs displayed complete killing in a larger proportion of isolates than the clinically approved AMP Polymyxin B, diminishing the risk of outgrowth of residual bacteria after antimicrobial treatment. Therefore, SET-M33 AMPs could potentially play a role in preventing biofilm-related infections, treating existing biofilm infections, and preventing the outgrowth of biofilm bacteria once treatment has stopped. The next step is to evaluate the efficacy of these AMPs, and particularly SET-M33L-PEG, in in vivo experiments.

## Figures and Tables

**Figure 1 antibiotics-14-00699-f001:**
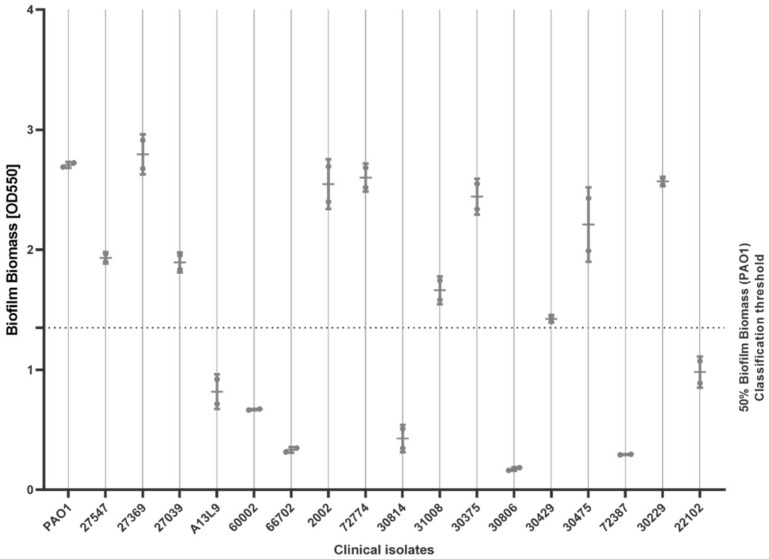
Differentiating between ‘weak’ and ‘strong’ biofilm-forming strains for 17 *P. aeruginosa* isolates using the crystal violet assay. Quantitative evaluation of biofilm formation for 17 clinical isolates using the crystal violet assay. A ’strong’ biofilm-forming isolate was defined as having OD550 values (crystal violet assay) that were ≥50% of the OD550 value obtained using the strong biofilm former and comparator strain PAO1.

**Table 1 antibiotics-14-00699-t001:** Antimicrobial susceptibility of 17 clinical *P. aeruginosa* isolates to nine antimicrobials.

Isolate	AMG		BL			PMX	CPN		FQ
Isolate	TOB	GEN	CAZ	PIP-TAZ	AZT	COL	MER	IMP	CIP
27547	S	IE	I	I	nt	S	S	I	I
A13L9	S	nt	I	nt	nt	nt	nt	nt	nt
60002	S	nt	I	I	nt	S	S	I	I
31008	S	IE	I	I	nt	S	S	I	I
22102	S	nt	I	I	nt	S	S	I	I
66702	S	nt	I	I	nt	nt	S	**R**	I
2002	S	**IE (R)**	I	I	nt	S	S	I	I
72774	S	**IE (R)**	I	I	nt	S	S	I	I
30814	S	IE	I	I	nt	S	S	I	**R**
30375	S	IE	I	I	nt	**R**	S	I	I
30229	S	IE	I	I	nt	S	S	**R**	**R**
27369	S	**IE (R)**	**R**	**R**	nt	S	S	I	**R**
30475	S	nt	**R**	**R**	I	S	**R**	**R**	**R**
27039	**R**	nt	**R**	**R**	**R**	nt	**R**	**R**	I
30806	**R**	nt	**R**	**R**	**R**	nt	**R**	**R**	**R**
30429	**R**	**IE (R)**	I	I	nt	S	**R**	**R**	**R**
72387	**R**	nt	**R**	**R**	I	S	**R**	**R**	**R**

**Table 2 antibiotics-14-00699-t002:** Minimum inhibitory concentration (MIC) and minimum bactericidal concentration (MBC) of SET-M33 AMPs and conventional antibiotics on 10 strong biofilm-forming *P. aeruginosa* isolates.

Isolate	Antimicrobial Compounds [μM]							
	Tobramycin		Ceftazidime		SET-M33L		SET-M33L-PEG	
	MIC	MBC	MIC	MBC	MIC	MBC	MIC	MBC
PAO1	4.3	8.5	3.7	3.7	2.7	10.9	2.7	10.6
27547	4.3	8.5	1.8	7.3	1.4	5.5	0.7	2.7
31008	4.3	17.1	3.7	14.6	10.9	21.9	5.3	21.2
2002	8.5	17.1	3.7	7.3	1.4	5.5	1.3	5.3
72774	8.5	17.1	3.7	7.3	1.4	5.5	1.3	5.3
30375	4.3	17.1	1.8	7.3	10.9	43.7	10.6	42.4
30229	4.3	17.1	7.3	29.3	5.5	21.9	1.3	5.3
*27369*	2.1	8.5	29.3	58.5	1.4	2.7	0.7	2.7
*30475*	4.3	17.1	58.5	234.2	0.3	1.4	0.3	0.7
*27039*	547.6	1095.2	58.5	117.1	5.5	10.9	10.6	21.2
*30429*	34.2	68.4	14.6	117.1	2.7	10.9	5.3	10.6

MICs and MBCs values were obtained according to EUCAST guidelines with slight modifications, as described in Section 2.4. The four isolates labeled in italics represent MDR strains that are resistant to three or more antimicrobial classes.

**Table 3 antibiotics-14-00699-t003:** Minimum and maximum minimum biofilm inhibitory concentration (MBIC) of SET-M33 AMPs and control compounds on 10 strong biofilm-forming *P. aeruginosa* isolates.

Isolate	Antimicrobial Compounds [μM]							
	Tobramycin		Polymyxin B		SET-M33L		SET-M33L-PEG	
	MBIC min	MBIC max	MBIC min	MBIC max	MBIC min	MBIC max	MBIC min	MBIC max
PAO1	4.3	8.6	1.6	3.3	2.7	5.5	2.7	5.3
27547	2.1	4.3	0.4	0.8	1.4	2.7	0.6	1.3
31008	8.6	17.1	0.8	1.6	5.5	10.9	5.3	10.6
2002	17.1	34.2	0.8	1.6	5.5	10.9	5.3	10.6
72774	17.1	34.2	0.8	1.6	5.5	10.9	5.3	10.6
30375	8.6	17.1	-	>106.3	10.9	21.8	10.6	21.2
30229	4.3	8.6	0.4	0.8	5.5	10.9	2.7	5.3
*27369*	4.3	8.6	0.4	0.8	2.7	5.5	1.3	2.6
*30475*	17.1	34.2	0.2	0.4	0.6	1.4	0.3	0.6
*27039*	136.8	273.8	1.6	3.3	5.5	10.9	10.6	21.2
*30429*	68.4	136.8	0.8	1.6	10.9	21.8	5.3	10.6

MBIC values were obtained the crystal violet methodology, as described in Section 2.5. The four isolates labeled in italics represent MDR strains that are resistant to three or more antimicrobial classes.

**Table 4 antibiotics-14-00699-t004:** Viability reduction in 10 strong biofilm-forming isolates of *P. aeruginosa* after treatment with SET-M33-AMPs and conventional antibiotics as MBIC.

Isolate	Untreated	Tobramycin				Polymyxin B				
PAO1	0	2.1	4.3	8.6	17.1	0.8	1.7	3.3	6.6	μM
	1 × 10^8^	1 × 10^9^	1 × 10^7^	1 × 10^4^	0.0	1 × 10^8^	1 × 10^8^	1 × 10^4^	1 × 10^3^	CFU/mL
27547	0	1.1	2.1	4.3	8.6	0.2	0.4	0.8	1.7	μM
	1 × 10^7^	1 × 10^7^	1 × 10^6^	1 × 10^5^	0.0	1 × 10^7^	1 × 10^6^	1 × 10^3^	0.0	CFU/mL
31008	0	4.3	8.6	17.1	34.2	0.4	0.8	1.7	3.3	μM
	1 × 10^9^	1 × 10^8^	1 × 10^8^	1 × 10^6^	1 × 10^3^	1 × 10^9^	1 × 10^9^	1 × 10^9^	1 × 10^8^	CFU/mL
2002	0	8.6	17.1	34.2	68.4	0.4	0.8	1.7	3.3	μM
	1 × 10^8^	1 × 10^7^	1 × 10^6^	0.0	0.0	1 × 10^8^	1 × 10^7^	1 × 10^4^	1 × 10^3^	CFU/mL
72774	0	8.6	17.1	34.2	68.4	0.4	0.8	1.7	3.3	μM
	1 × 10^8^	1 × 10^7^	1 × 10^6^	0.0	0.0	1 × 10^8^	1 × 10^7^	1 × 10^4^	1 × 10^3^	CFU/mL
30375	0	4.3	8.6	17.1	34.2	26.6	53.2	106.4	212.7	μM
	1 × 10^7^	1 × 10^6^	1 × 10^4^	1 × 10^4^	0.0	1 × 10^7^	1 × 10^7^	1 × 10^6^	1 × 10^5^	CFU/mL
30229	0	2.1	4.3	8.6	17.1	0.4	0.8	1.7	3.3	μM
	1 × 10^8^	1 × 10^7^	1 × 10^7^	1 × 10^5^	1 × 10^4^	1 × 10^7^	1 × 10^7^	1 × 10^7^	1 × 10^5^	CFU/mL
*27369*	0	2.1	4.3	8.6	17.1	0.2	0.4	0.8	1.7	μM
	1 × 10^7^	1 × 10^7^	1 × 10^7^	1 × 10^5^	1 × 10^2^	1 × 10^6^	1 × 10^6^	1 × 10^6^	1 × 10^3^	CFU/mL
*30475*	0	8.6	17.1	34.2	68.4	0.2	0.4	0.8	1.7	μM
	1 × 10^7^	1 × 10^6^	0.0	0.0	0.0	1 × 10^7^	1 × 10^6^	1 × 10^2^	0.0	CFU/mL
*27039*	0	68.4	136.9	273.3	547.6	0.8	1.7	3.3	6.6	μM
	1 × 10^6^	1 × 10^7^	1 × 10^6^	1 × 10^4^	0.0	1 × 10^6^	1 × 10^5^	1 × 10^2^	0.0	CFU/mL
*30429*	0	34.2	68.4	136.9	273.3	0.4	0.8	1.7	3.3	μM
	1 × 10^8^	1 × 10^7^	1 × 10^5^	0.0	0.0	1 × 10^8^	1 × 10^7^	1 × 10^7^	1 × 10^5^	CFU/mL
Isolate	**Untreated**	**SET-M33L**				**SET-M33L-PEG**				
PAO1	0	1.4	2.7	5.5	10.9	1.3	2.7	5.3	10.6	μM
	1 × 10^8^	1 × 10^8^	1 × 10^8^	1 × 10^5^	0.0	1 × 10^8^	1 × 10^8^	1 × 10^5^	0.0	CFU/mL
27547	0	0.7	1.4	2.7	5.5	0.3	0.7	1.3	2.7	μM
	1 × 10^7^	1 × 10^8^	1 × 10^6^	1 × 10^4^	1 × 10^3^	1 × 10^7^	1 × 10^7^	1 × 10^4^	1 × 10^3^	CFU/mL
31008	0	2.7	5.5	10.9	21.8	2.7	5.3	10.6	21.2	μM
	1 × 10^9^	1 × 10^8^	1 × 10^8^	1 × 10^6^	1 × 10^3^	1 × 10^8^	1 × 10^8^	1 × 10^5^	1 × 10^3^	CFU/mL
2002	0	2.7	5.5	10.9	21.8	2.7	5.3	10.6	21.2	μM
	1 × 10^8^	1 × 10^8^	1 × 10^8^	1 × 10^5^	0.0	1 × 10^9^	1 × 10^8^	1 × 10^4^	0.0	CFU/mL
72774	0	2.7	5.5	10.9	21.8	2.7	5.3	10.6	21.2	μM
	1 × 10^8^	1 × 10^8^	1 × 10^8^	1 × 10^5^	0.0	1 × 10^9^	1 × 10^8^	1 × 10^4^	0.0	CFU/mL
30375	0	5.5	10.9	21.8	43.7	5.3	10.6	21.2	43.7	μM
	1 × 10^7^	1 × 10^7^	1 × 10^6^	1 × 10^3^	0.0	1 × 10^7^	1 × 10^7^	1 × 10^5^	0.0	CFU/mL
30229	0	2.7	5.5	10.9	21.8	1.3	2.7	5.3	10.6	μM
	1 × 10^8^	1 × 10^8^	1 × 10^8^	1 × 10^7^	1 × 10^4^	1 × 10^8^	1 × 10^8^	1 × 10^8^	1 × 10^5^	CFU/mL
*27369*	0	1.4	2.7	5.5	10.9	0.7	1.3	2.7	5.3	μM
	1 × 10^7^	1 × 10^8^	1 × 10^8^	1 × 10^7^	1 × 10^2^	1 × 10^7^	1 × 10^7^	1 × 10^4^	1 × 10^3^	CFU/mL
*30475*	0	0.3	0.7	1.4	2.7	0.2	0.3	0.7	1.3	μM
	1 × 10^7^	1 × 10^7^	1 × 10^5^	1 × 10^3^	1 × 10^2^	1 × 10^7^	1 × 10^6^	1 × 10^5^	1 × 10^3^	CFU/mL
*27039*	0	2.7	5.5	10.9	21.8	5.3	10.6	21.2	43.7	μM
	1 × 10^6^	1 × 10^6^	1 × 10^6^	1 × 10^6^	1 × 10^3^	1 × 10^6^	1 × 10^6^	1 × 10^4^	0.0	CFU/mL
*30429*	0	5.5	10.9	21.8	43.7	2.7	5.3	10.6	21.2	μM
	1 × 10^8^	1 × 10^8^	1 × 10^5^	1 × 10^3^	1 × 10^3^	1 × 10^8^	1 × 10^7^	1 × 10^2^	0.0	CFU/mL

Viability was measured through colony counting by plating a pre-treated biofilm with antimicrobial concentrations ranging from 0.5xMBIC to 2xMBIC. The four isolates labeled in italics represent MDR strains, resistant to three or more antimicrobial classes.

**Table 5 antibiotics-14-00699-t005:** FIC indices for combinations of SET-M33 AMPs with the conventional antimicrobials tobramycin and ceftazidime on 10 strong biofilm-forming isolates of *P. aeruginosa*.

Isolate	SET-M33L		SET-M33L-PEG	
	Tobramycin	Ceftazidime	Tobramycin	Ceftazidime
PAO1	1	0.8	1	0.8
27547	1	1.5	1	1.5
31008	1	0.8	1	0.8
2002	1	1	0.8	1
72774	1	1	0.8	1
30375	1.3	1.5	1.3	1
30229	0.8	0.4	1.3	1.1
*27369*	1	1.1	1.5	1.1
*30475*	2.3	2.1	2.3	2.1
*27039*	1.1	0.6	1.1	0.4
*30429*	1	0.8	0.8	0.6

Combined MICs values were obtained according to EUCAST guidelines with slight modifications, as described in Section 4.7. FIC indices were calculated as follows: FIC index = FIC (A) + FIC (B), where FIC (A) = MIC of drug A in combination/MIC of drug A alone; FIC (B) = MIC of drug B in combination/MIC of drug B alone. The four isolates labeled in italics represent MDR strains that are resistant to three or more antimicrobial classes.

**Table 6 antibiotics-14-00699-t006:** FBC indices for combinations of SET-M33 AMPs with the conventional antimicrobials tobramycin and ceftazidime on 10 strong biofilm-forming isolates of *P. aeruginosa*.

Isolate	SET-M33L		SET-M33L-PEG	
	Tobramycin	Ceftazidime	Tobramycin	Ceftazidime
PAO1	0.8	0.8	0.8	0.8
27547	0.5	0.6	0.6	0.5
31008	0.6	0.4	0.5	0.4
2002	0.6	0.6	0.6	0.6
72774	0.6	0.6	0.6	0.6
30375	0.3	0.3	0.3	0.4
30229	0.3	0.1	0.6	0.4
*27369*	0.8	1.1	0.5	0.6
*30475*	0.6	0.6	1.1	1.1
*27039*	0.6	0.6	0.6	0.4
** *30429* **	0.6	0.2	0.8	0.4

Combined MBCs values were obtained according to EUCAST guidelines with slight modifications, as described in Section 4.7. FBC indices were calculated as follows: FBC index = FBC (A) + FBC (B), where FBC (A) = MIC of drug A in combination/MBC of drug A alone; FBC (B) = MBC of drug B in combination/MBC of drug B alone. The four isolates labeled in italics represent MDR strains that are resistant to three or more antimicrobial classes.

## Data Availability

Data are available upon application to the corresponding author in conjunction with a signed data access agreement.

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
