# Peer review of "Antimicrobial Peptides SET-M33L and SET-M33L-PEG Are Promising Agents Against Strong Biofilm-Forming P. aeruginosa, Including Multidrug-Resistant Isolates"

_antibiotics, 2025, doi:10.3390/antibiotics14070699_

Round 1

Reviewer 1 Report

Comments and Suggestions for Authors

Summary: This study evaluates the antimicrobial and antibiofilm activity of SET-M33L and SET-M33L-PEG, two synthetic antimicrobial peptides, against strong biofilm-forming and multidrug-resistant (MDR) Pseudomonas aeruginosa isolates commonly associated with chronic pulmonary diseases. The findings indicate that these peptides exhibit significantly lower MICs and MBCs compared to conventional antibiotics like tobramycin and ceftazidime, especially in MDR strains. SET-M33L-PEG was generally more effective than its non-pegylated counterpart, showing enhanced biofilm inhibition. The study also found synergistic bactericidal effects when these peptides were combined with tobramycin and ceftazidime, potentially enhancing therapeutic outcomes. While the peptides outperformed tobramycin in biofilm inhibition, polymyxin B remained slightly superior in some cases. These findings suggest SET-M33 AMPs as promising alternatives for treating resistant P. aeruginosa infections, reducing biofilm formation, and preventing bacterial regrowth post-treatment. Further in vivo studies are needed to validate their clinical potential.

General Comments: The study confirmed lower MICs and MBCs of SET-M33L and SET-M33L-PEG compared to tobramycin and ceftazidime, particularly in MDR strains; demonstrated biofilm inhibition, outperforming tobramycin but performing slightly behind polymyxin B and showed synergy (FBC < 0.5) with tobramycin and ceftazidime in most isolates, reinforcing their potential as combinatory treatments. Additionally, SET-M33L-PEG was found to be more effective than SET-M33L, suggesting PEGylation as a viable strategy for improving AMP activity against biofilm formation.

The MBIC assay measures the prevention of biofilm formation over 24 hours. The study did not evaluate the peptides' ability to destroy or eradicate a pre-formed, mature biofilm, which is a more clinically relevant and difficult challenge.

The paper reports an interesting finding where AMP-antibiotic combinations were additive for inhibition (FIC) but synergistic for killing (FBC). However, it does not investigate the underlying mechanism for this powerful synergy.

The paper proposes the AMPs as "promising agents" but does not include any data on their cytotoxicity against human cells (e.g., lung epithelial cells). While citing previous work on toxicity, presenting concurrent toxicity data would significantly strengthen the argument for their clinical potential.

Specific Comments:

  • Some sentence structures are dense, making them slightly difficult to read. Breaking them into shorter sentences or bullet points could improve clarity.
  • There are several instances of inconsistent spacing after commas or periods throughout the document.
  • The synergy results could benefit from a more detailed discussion of potential mechanisms (e.g., why FIC and FBC results differ in some cases).
  • It might help to provide more visual representation of trends—such as scatter plots or comparative graphs for MICs and MBCs across AMPs vs. conventional antibiotics.

Reviewer 2 Report

Comments and Suggestions for Authors

Reviewer Comments

  • Abstract – Line 20-21: All the scientific names should be in italics. Please change to italics.
  • Introduction: The work is about antimicrobial activity of the synthetic peptides against Pseudomonas aeruginosa however the introduction does not give enough information about the pathogen in question. Please revise the introduction accordingly.
  • Was ethical clearance taken for the work? The ethical clearance statement along with the number should be included in the manuscript.
  • Results - Fig 1A: The distinction of high and low biofilm is not obvious in the microscopic images, also it is qualitative data. A quantification of the images could be done. The quantification of biofilm by CV assay however is fine in Fig 1B.
  • Was scanning electron microscopy or confocal microscopy done? If so, please include as this will leverage the antimicrobial work.

Reviewer 3 Report

Comments and Suggestions for Authors

In this manuscript, the authors evaluate the efficacy of the antimicrobial peptides SET-M33L and SET-M33L-PEG against strong biofilm-forming P. aeruginosa. The manuscript is generally well structured; however, it lacks in-depth discussion. Therefore, the manuscript cannot be considered for publication until the following issues are addressed:

The discussion section should be expanded to provide deeper insights into the mechanisms underlying the additive or synergistic effects observed when SET-M33L and SET-M33L-PEG are used in combination with other antimicrobials.

A schematic representation of both SET-M33L and SET-M33L-PEG is recommended to help readers better understand their structures and potential mechanisms of action.

Several editorial and terminology issues should be corrected:

(1) The abbreviation “MDR” should not be used in the title; the full term should be written out.

(2) In line 23, “pegylated” should be capitalized as “PEGylated.”

(3) The phrase “pegylated SET-M33L-PEG” is redundant and confusing, as SET-M33L-PEG is already described as PEGylated.

Round 2

Reviewer 3 Report

Comments and Suggestions for Authors

The manuscript can be accepted at the present form.